# Determining the Development Strategy and Suited Adoption Paths for the Core Competence of Shared Decision-Making Tasks through the SAA-NRM Approach

**DOI:** 10.3390/ijerph192013310

**Published:** 2022-10-15

**Authors:** Shan-Fu Yu, Hui-Ting Wang, Meng-Wei Chang, Tien-Tsai Cheng, Jia-Feng Chen, Chia-Li Lin, Hsing-Tse Yu

**Affiliations:** 1Division of Rheumatology, Allergy, and Immunology, Department of Internal Medicine, Kaohsiung Chang Gung Memorial Hospital, Kaohsiung 833, Taiwan; 2Division of Rheumatology, Allergy, and Immunology, Department of Internal Medicine, Chiayi Chang Gung Memorial Hospital, Chiayi 613, Taiwan; 3School of Medicine, College of Medicine, Chang Gung University, Tayouan 333, Taiwan; 4Graduate Institute of Adult Education, National Kaohsiung Normal University, Kaohsiung 802, Taiwan; 5Department of Emergency Medicine, Kaohsiung Chang Gung Memorial Hospital, Kaohsiung 833, Taiwan; 6Department of International Business, Ming Chuan University, Taipei 111, Taiwan; 7Department of Obstetrics and Gynecology, Taipei Chang Gung Memorial Hospital, Taipei 105, Taiwan

**Keywords:** shared decision making, physician, competency, DEMATEL, SAA-NRM

## Abstract

Shared decision making (SDM) is an interactive process that involves patients and their healthcare professionals reaching joint decisions about medical care through negotiation. As the initiators of medical decision-making in daily routine, physicians should be aware of and concerned about the SDM process. Thus, professional competency development for SDM has become increasingly critical for physicians’ training. Therefore, this study investigates the professional competency and the important competency development aspects/criteria of SDM tasks through expert interviews and literature research. The study adopts the SAA (satisfaction-attention analysis) method to assess the status of competency development aspects/criteria and determine the NRM (network relation map) based on the DEMATEL (decision-making trial and evaluation laboratory) technique. The results demonstrate that the CE (concept and evaluation) aspect is the dominant aspect, and the CR (communication and relationship) aspect is the aspect being dominated. The CE aspect influences the aspects of SP (skill and practice), JM (joint information and decision making) and CR, and the SP aspect affects the aspects of JM and CR. Then, the JM aspect affects the CR aspect. The study also suggests suitable adoption paths of competency development for SDM tasks using the NRM approach. It provides recommendations and strategic directions for SDM competency development and sustainable training programs.

## 1. Introduction

Shared decision-making (SDM) is a flexible way to incorporate paternalistic and informative models into the modern concept of patient-centered care. By listening to patients, proposing options, and keeping their choice, physicians’ expertise is relevant to biomedical and relational skills [1]. Patients’ involvement in this decision-making process may affect their satisfaction and quality of life, and may lead to more significant adherence to treatment, and improve the doctor-patient relationship [2,3]. This collaborative process helps patients understand the risks, benefits, and consequences of different options and make appropriate medical decisions by debating, exchanging information, and respecting patient autonomy [4]. Uncertainty surrounding the evolving COVID-19 pandemic is exceptionally high. Individuals consider the specific context when making COVID-19 related decisions [5]. Integrating SDM concepts with VALUE (value family statements, acknowledge emotions, listen, understand the patient as a person, elicit questions) and PLACE (prepare with intention, listen intently and completely, agree on what matters most, connect with the patient’s story, explore emotional cues) processes may improve the quality of care using telehealth during the pandemic [6].

Furthermore, SDM is particularly helpful in complex situations where multiple treatment choices exist, such as the decisions needed when dealing with a cancer diagnosis [7]. The diagnosis and treatment of cancer require numerous high-risk decisions within limited time windows, and the evidence is often insufficient. As a result, patients with cancer often need more support in the decision-making process. Let us assume a theoretical case report: Ms. Wang is a 30-year-old woman without any medical problems. She was recently diagnosed with invasive, non-metastatic breast cancers. The therapy for these cancers usually consists of chemotherapy, radiotherapy, or hormone therapy followed by surgery. She was offered a surgical choice between breast conservation surgery and a total mastectomy. She was notified that the two surgical procedures both result in disease-free survival and have an equal overall survival rate. She was shocked by this choice and became worried. Although given good information, she listened to the advice and felt steered toward having a total breast conservation surgery as the “fewer postsurgical complications, better quality of life, sexual health, and cosmetic consideration” option. She had no discomfort except skin hyperpigmentation and an itchy sensation of the chest wall during the radiotherapy, and her breasts became much smaller. Three years later, the local recurrence of breast cancer necessitated a total mastectomy. She became aware of a higher recurrence of breast cancer in women younger than 40. She felt regret and thought that her decision might have been different if she had been provided with more information and a chance to express her intense wish to avoid the recurrence of breast cancer. In this case, a decision must be made on whether breast conservation surgery suits the patient. There is often a disparity of information between doctors and patients in clinical settings. Clinicians prefer to make decisions informed by medical knowledge and evidence, while patients are sensitive to the strict expectations of recovery and quality of life [8]. Where possible, her doctor had an excellent SDM dialogue with Ms. Wang before surgery. Ms. Wang also had sufficient knowledge to understand the proposed treatment and build a sustainable opinion. It might have been possible to avoid her decision regret and improve her treatment satisfaction based on the goal of patient-centered care [9]. To successfully implement SDM, it is important not only the evaluation of physicians’ professionalism but also to consider the patient’s training and support the patient’s medical literacy to enable their engagement in healthcare [10,11,12]. Health literacy, especially the ability to access information and communicate interactively, plays a prominent role in implementing SDM with cancer patients [13].

Competence in medicine can be described as possessing the knowledge, skills, and experience to meet the demands of the role of a medical professional [14]. A “core competency” is a broader structure of two or more competencies. It is difficult for one person to master multiple capabilities simultaneously. To effectively initiate the application of competencies, it is helpful to focus on a few critical capabilities and enhance them gradually [15]. Scholars have proposed important conceptual work on SDM, but significant challenges remain for its implementation in clinical practice. In most clinical conditions, the clinician is always the initiator of medical decisions and should be familiar with participating in the SDM process. Identifying physicians’ competencies, developing tools to evaluate key capabilities, and providing interventions to enhance these capabilities may also be necessary. However, most healthcare professionals enter the healthcare setting without good SDM skills. Many practicing physicians may be unfamiliar with the SDM concepts [16]. Some studies and guidelines suggest that health providers are more likely to apply SDM if well trained [17,18]. Most past studies on the SDM implementation process focus on a simple model or framework, such as the three-talk model [19,20] or the SHARE (seek, help, assess, reach, evaluate) approach [21], while ignoring the interaction between different influencing factors. Therefore, the question is how to segment core competencies into elements to facilitate the ability development of physicians to perform SDM tasks adequately. The decision-making trial and evaluation laboratory (DEMATEL) approach is suitable for solving this problem [22]. The DEMATEL approach helps to collect data to form structural models and analyze the causal relationships of subsystems via cause-and-effect diagrams.

To enforce SDM in the daily routine at clinics and hospitals in Taiwan, a national SDM program was initiated in spring 2016, using various methods to implement it, including the development of decision aids for patients, the building of SDM platforms, and the implementation of SDM in clinical settings [23]. SDM is seen as a new concept for health professionals, patients, and medical society as a whole in Taiwan. Prior studies on SDM have been broadly divided into key components/process steps, barriers and facilitators, SDM scales, content of curriculum training, and patient decision aid application [4,6,8,24,25]. There is little evidence of the priorities of physicians’ core competencies required to perform successful SDM. Therefore, this study used an online questionnaire to investigate the prioritization and causality of competencies necessary for SDM implementation. Through literature reviews and expert interviews, we define the competence-driving forces of SDM tasks, including four aspects and 16 evaluation criteria. This study initially presents the SAA (satisfaction-attention analysis) approach, which incorporates the performance (satisfaction index, SI) and perspective of physicians (attention index, AI) to evaluate these critical core competencies. Furthermore, this study utilizes the NRM (network relation map) approach to establish the influence relation structure of the competence-driving force and identify effective adoption strategies for SDM development. Ultimately, this study integrates the SAA-NRM techniques to determine suitable improvement planning for medical education on SDM implementation. We can adopt the empirical results to propose a new SDM teaching and training approach.

## 2. Materials and Methods

This research adopts the analytic process (e.g., SAA technique, NRM technique, and SAA-NRM approach) and six analytical steps, as shown in Figure 1. First, we identify the critical decision problem of the competency development system. Next, we define the core competencies (aspects/criteria) of SDM tasks by expert interviews and literature reviews. Then, we evaluate the satisfaction index and attention index for each aspect/criterion and apply the SAA technique to assess the satisfaction and attention status in the third step. After that, this study analyzes the NRM approach’s network relation structure for the SDM adoption strategies and determines the dominant aspects/criteria. In the fifth step, the study integrates the results of the SAA and NRM technique to present an adoption strategy for SDM tasks. Ultimately, we specify the suitable adoption paths by the aspects/criteria rank for satisfaction and attention. This study uses Microsoft Office Excel to establish the SAA approach and Matlab to calculate the NRM approach.

### 2.1. The Literature Reviews and Expert Interviews

The literature search strategy of articles in our study was performed using PubMed and ScienceDirect databases with Boolean operators to construct physicians’ SDM competency indicators. This study used the keywords “shared decision making”, “competency”, “competencies”, “knowledge”, “skill”, and “attitude” to review the literature and summarized the concepts of SDM, SDM assessment tools/scales, and essential elements in the SDM process. The study also adopts the MECE (Mutually Exclusive Collectively Exhaustive) concept to establish the evaluation system of service/decision process. The MECE principle has three stages, including measuring independence, importance, and completeness [26]. After analyzing the existing literature and related works, we carried out expert interviews to identify these indicators of SDM competence among physicians. Four experienced experts in the Shared Decision-Making Group at the Centre for Quality Management, Kaohsiung Chang Gung Memorial Hospital participated in this study. Several potential aspects/criteria can be selected based on the literature collection and analysis. The results are then used as the central point for discussion. After combining interviews with experts and literature research, the content is organized into the questionnaire design using the subsequent definitions. There are four aspects, including conception and evaluation (CE), skill and practice (SP), communication and relationship (CR), and joint information and decision making (JM). Each aspect comprised four criteria and related descriptions, as shown in Table 1.

#### 2.1.1. The Conception and Evaluation (CE) Aspect

SDM is an interactive process in which health professionals discuss preference-sensitive decisions with patients in a joint and negotiated form so patients can choose options for integrating their values and preferences into evidence-based medicine [4,27]. It can build new associations between people and professionals based on partnerships [28]. SDM attempts to improve information asymmetry between healthcare professionals and patients. It is appropriate to initiate SDM for patients with uncertain prognosis or when there is difficulty determining between options [8]. Physicians agree to use an evidence-based approach for SDM implementation and have basic knowledge of evidence-based medicine as a specific element of the SDM process [29]. In short, conception and evaluation are core competencies needed to use SDM. We summarize four evaluation criteria associated with the CE (conception and evaluation) aspect, including the concept (CE1), importance and value (CE2) of SDM, awareness and evaluation (CE3), and evidence-based knowledge (CE4), as shown in Table 1.

#### 2.1.2. The Skill and Practice (SP) Aspect

How to execute SDM? The SHARE approach and the three-talk model are two of the most common frameworks that can help physicians develop SDM competencies [19,20,21]. With the diversity of complex diseases and expanding treatment options, treatment decisions will become very delicate and require SDM. The SHARE approach and three-talk model combined with motivational interviews and brief counseling may help adolescents make informed decisions based on preferences for reducing potential health hazards [30]. Choosing a decision aid is helpful to the patient, as such assistance can ameliorate patient knowledge, quality, and efficiency of care [31,32,33]. The SDM handles power asymmetries and restores autonomy and agency when needed [34]. SDM lets physicians use evidence-based information while positioning the patient (and, where suitable, family members) at the center of clinical decisions [35]. So, training in SDM skills is part of evidence-based practice. This study outlines four evaluation criteria concerning the SP (skill and practice) aspect, including decision-making step (SP1), evidence-based medicine skills (SP2), use of patient aids (SP3), and engagement skills (SP4), as shown in Table 1.

#### 2.1.3. The Communication and Relationship (CR) Aspect

Two intertwined processes of the main SDM work are communication and collaboration. This work needs to build a good relationship, no matter how short-term, as SDM supposes that all involved parties are communicating sincerely [34]. A conversational model of risk communication facilitates high-quality decisions in cancer care because it addresses open spaces to discover links between clinical evidence and patient values while permitting a satisfactory level of participation from all parties [36]. The teach-back method is a practical communication instrument that can be used to improve SDM and patient safety [37]. Its utility in patient care has been seen in many clinical scenarios. Most SDM models focus on doctor-patient dyads. Nevertheless, the Interprofessional SDM model designed by Legare and colleagues emphasizes that team service staff from other healthcare disciplines can play a critical role [38]. Teaming with multi-disciplinary members can be especially useful when encouraging the adoption of a patient decision aid [39]. This study summarizes four evaluation criteria associated with the CR (communication and relationship) aspect, including establishment of the doctor-patient relationship (CR1), communication in verbal and non-verbal forms (CR2), reply and teach (CR3), and teamwork and collaboration (CR4), as shown in Table 1.

#### 2.1.4. The Joint Information and Decision Making (JM) Aspect

Physicians must notify the patient that a decision needs to be made and clarify the patient’s fundamental role in decision-making [40]. Flexibility in how doctors structure the decision-making process is essential to respect individual differences in patient preferences [41]. Healthcare professionals share evidence-based information concerning available options and discuss each option’s benefits, risks, costs, and uncertainties while permitting sufficient time for inquiries [42]. Explaining the association between risk factors and individualized risk estimates may help patients understand and reflect on these [43]. SDM can usually be done in a “distributed” fashion over multiple visits [27]. Patients can reconsider decisions if available treatment options do not produce the desired health outcomes [8]. We outline four evaluation criteria concerning the JM (joint information and decision making) aspect, including defining decision-making needs (JM1), sharing information (JM2), confirming decision-making (JM3), and following up on medical decisions (JM4), as shown in Table 1.

**Table 1 ijerph-19-13310-t001:** The definitions of aspects/criteria for competency evaluation of SDM tasks.

Aspects/criteria	Descriptions	References
Concept and Evaluation (CE)		
Concept of SDM (CE1)	Recognize the definition of SDM and understand the goal of SDM.	[4,27,44]
Importance and value of SDM (CE2)	Comprehend the importance of SDM and agree with the value of SDM.	[27,45]
Awareness and evaluation (CE3)	Ability to distinguish scenarios that are suitable and not suitable for SDM.	[8,27,46]
Evidence-based knowledge (CE4)	Using evidence-based knowledge support physicians’ ability to conduct the SDM process.	[29,47]
Skill and Practice (SP)		
Decision-making step (SP1)	Have sufficient capability in medical consultation to implement the steps required for SDM.	[19,20,21] Expert
Evidence-based medicine skills (SP2)	Familiarity with the skills of reviewing evidence from the literature and applying them appropriately to support decision-making in patient care.	[29,47] Expert
Use of patient aids (SP3)	Know the decision aid platform, and make good use of patient decision aids to assist medical decision-making.	[33,39,47] Expert
Engagement skills (SP4)	Encourage patient participation, guide patients to express personal opinions, improve patient self-efficacy, and respect for autonomy.	[34,47] Expert
Communication and Relationship (CR)		
Establishment of the doctor-patient relationship (CR1)	Learn to communicate effectively and establish a good doctor-patient relationship with patients with different socioeconomic cultures, backgrounds, and personality traits and their families, and give empathic responses.	[34,48,49]
Communication in verbal and non-verbal forms (CR2)	Avoid using specialized terms that patients do not easily understand, try to use simple words and phrases familiar to patients, fully allow patients to express their wishes, provide an appropriate amount of information, clarify myths and emphasize a crucial information.	[49,50,51]
Reply and teach (CR3)	Select the proper time and method of reply and teach, guide the patient to repeat the teaching content or show their understanding, and confirm the correctness of the patient’s reply and teach.	[52,53] Expert
Teamwork and collaboration(CR4)	Learn to establish effective communication with team members, understand how to collaborate with other professionals, jointly evaluate and coordinate to improve the patient-centered care.	[38,54]
Joint Information and Decision Making (JM)		
Defining decision-making needs (JM1)	Integrate the patient’s clinical problems, respect the patient’s values and preferences, and develop appropriate multiple-choice options through high-quality evidence appraisal.	[40,41,47]
Sharing information (JM2)	Provide all necessary information, the advantages, and disadvantages, analyze the risks of various treatment options (including health insurance/self-payment), clarify the patient’s understanding and expectations, and understand the patient’s concerns and fears.	[21,42,48] Expert
Confirming decision-making (JM3)	The decision-making process remains flexible, invites people who can support the decision to accompany the patient, give the patient enough time to think about the decision, and ensure that the patient has a complete understanding of the treatment options, check the decision-making process, and make a record of the what and why of shared decisions.	[19,27,43]
Following up on medical decisions (JM4)	Arrange follow-up to evaluate the treatment effect of the decision, implementation status, patient satisfaction, and changes in anxiety, and to determine whether the initial decision was right.	[8,19,44] Expert

### 2.2. The Research Design and the Reliability Analysis

This study was approved by the local Institutional Review Board of Chang Gung Memorial Hospital (202200716B0, 202200716B0C501) and was performed according to the principles of the Declaration of Helsinki. The aspects and criteria were used to construct the questionnaires through literature reviews and experts’ interviews with the SDM group of the hospital. An 11-point Likert scale (0–10) was used to collect physicians’ satisfaction and attention to aspects/criteria. Data was collected through online questionnaires.

### 2.3. The SAA Approach

The SAA approach exploring the SI and AI was conducted, and the surveyed data was normalized into equivalent calculating scales. In the normalized procedure, these aspects can be separated into four quadrants: (1) the first quadrant denotes the high satisfaction and high attention level (H, H), (2) the second quadrant denotes the low satisfaction and high attention level (L, H), (3) the third quadrant denotes the low satisfaction and low attention level (L, L), and (4) the fourth quadrant denotes the high satisfaction and low attention level (H, L). 

### 2.4. The NRM Analysis Based on the DEMATEL Approach

The NRM map for the core competencies of SDM development is built by the DEMATEL technique. While physicians join patients in making decisions, many criteria may be considered. The most common issue they encounter is that these aspects impact each other. Therefore, before learning SDM issues, it is necessary to know the essential criteria and make helpful capability developments to improve overall satisfaction. When a decision-maker needs to enhance many aspects, the best practice to manage this is to define the aspects that have the most significant impact on others and strengthen them.

Several current studies have adopted the DEMATEL technique to evaluate complicated issues, such as user interface analysis [55], intertwined evaluation in e-learning programs through a hybrid multiple criteria decision making (MCDM) model [56], building airline safety management system [57], evaluating value-created systems for science (technology) parks [58], selecting vehicle telematics system [59], improving the performance in a matrix organization [60], evaluation of design delay factors by importance-satisfaction analysis and NRM [61], selecting the model for digital music service platforms [62], the analysis of the environmental sustainability challenges in the Indian automobile industry [63], identifying the sustainable development strategies for industrial tourism via innovation opportunity analysis-NRM approach [64], determining critical performance criteria for hospital management by the double hierarchy hesitant fuzzy linguistic term sets (DHHFL)—DEMATEL method [65], the analysis of Med-tech industry entry strategy during pandemic [66], building the digital transformation strategies for the Med-Tech Enterprises by acquisition-importance analysis-NRM approach [67], identifying the critical success factors of SDM [8], investigation of factors impeding the dissemination of medical information standards [68], assessment of urban sustainable adoption strategies and common suited paths [69], planning urban revitalization and regional development strategies [70], analyzing the driving factors of urban music festival tourism and the strategies for service development [71], exploring the cause of gas explosion accidents by the DEMATEL-ISM method [72], evaluating the risk analysis of maritime accidents using the DEMATEL and ANP technique [73], and addressing the classifier selection problem in assistive technology adoption for patients with dementia by integrating the IF-DEMATEL and TOPSIS methods [74]. 

The DEMATEL technique is divided into five stages in this study: (1) estimate the original average matrix, (2) compute the direct influence matrix, (3) determine the indirect influence matrix, (4) count the full influence matrix, and (5) establish the NRM (network relation map).

Estimate the original average matrix

Respondents were asked to denote the impact they thought each aspect had on the other aspects on a scale from 0 to 4. “0” represents no influence, whereas “4” indicates “extreme influence” between aspects/criteria. “1”, “2”, and “3” represent “low influence”, “medium influence” and “high influence” respectively.

2.Compute the direct influence matrix

We presented the “original average influence matrix” (***A***) through Equations (1) and (2) and got the “direct influence matrix” (***D***). All diagonal items of ***D*** are 0, and the sum of a row is at a maximum of 1. We then set up by counting up rows and columns.
(1)D=sA, s>0
where
(2)$$s=\underset{i,j}{\min}\;[1/\underset{1\leq i\leq n}{\max}\sum_{j=1}^{n}a_{ij},1/\underset{1\leq j\leq n}{\max}\sum_{i=1}^{n}a_{ij}],i,j=1,2,\ldots ,n$$
and $\underset{m\to \infty }{\lim}\;D^{m}={\left[0\right]}_{n\times n}$, where $D={\left[x_{ij}\right]}_{n\times n}$, when $0<\sum_{j=1}^{n}x_{ij}\leq 1\;$ or $0<\sum_{i=1}^{n}x_{ij}\leq 1$, and at least one $\sum_{j=1}^{n}x_{ij}$ or $\sum_{i=1}^{n}x_{ij}$ equals one, but not all. So, we can guarantee $\underset{m\to \infty }{\lim}\;D^{m}={\left[0\right]}_{n\times n}$.

3.Determine the indirect influence matrix

The indirect influence matrix can originate from Equation (3).
(3)$$ID=\sum_{i=2}^{\infty }D^{i}=D^{2}{\left(I-D\right)}^{-1}$$

4.Count the full influence matrix

The total influence matrix ***T*** can originate from Equation (4) or Equation (5). The total influence matrix ***T*** contains multiple components, as demonstrated in Equation (6). The sum of row value is {d}, and the sum of column value is {r}; the sum of row value added to column value is {$d_{i}+r_{i}$}, representing the total influence of matrix ***T***. If the sum of row value plus column value {$d_{i}+r_{i}$} is large, the correlation of the aspect or criterion is high. The sum of row value minus column value is {$d_{i}-r_{i}$}, displaying the net influence relationship. If $d_{i}-r_{i}$ > 0, it indicates that the extent of affecting others is more powerful than the extent of being affected.
(4)$$T=D+ID=\sum_{i=1}^{\infty }D^{i}$$
(5)$$T=\sum_{i=1}^{\infty }D^{i}=D{\left(I-D\right)}^{-1}$$
(6)$$T=[t_{ij}],\;\;i,j\in \{1,2,\ldots ,n\}$$
(7)$$d=d_{n\times 1}=[\sum_{j=1}^{n}t_{ij}{]}_{n\times 1}=(d_{1},\ldots ,d_{i},\ldots ,d_{n})$$
(8)$$r=r_{n\times 1}=[\sum_{i=1}^{n}t_{ij}{{]}^{\prime }}_{1\times n}=(r_{1},\ldots ,r_{j},\ldots ,r_{n})$$

5.Establish the NRM (network relation map)

The net full influence matrix, ***T******_net_***, is defined by Equation (9).
(9)$$T_{net}=[t_{ij}-t_{ji}],i,j\in \{1,2,\ldots ,n\}$$

All diagonal items of the matrix are 0. In other terms, the matrix includes a precisely upper and lower triangular matrix. Furthermore, although the upper and lower triangular matrices have equal values, they have opposite symbols. This property aids us; we only need to choose one of the triangular matrices. 

### 2.5. The Analysis of the SAA-NRM Approach

The analytic procedure of SAA-NRM consists of two stages. The first stage is the SAA method, and the second stage is the NRM technique. The SAA analysis defines the aspects/criteria state of satisfaction and attention index for competency development; the SAA analysis can help decision-makers to recognize aspects that should be developed when the standard satisfied index is lower than the average satisfied index. The SAA-NRM approach defines the aspects/criteria that should be improved through the SAA analysis and the development path based on the NRM approach. We can then determine the preferred strategy by integrating the findings of the SAA and NRM approaches.

## 3. Results

This study uses the SAA-NRM approach to evaluate competency development strategies for four aspects. We survey the satisfaction index and attention index through the questionnaire. It identifies the aspects that should be improved through the SAA approach and determines the suited development strategies using the NRM approach. The SAA and NRM methods are incorporated to define the preferred development strategy and a proper improvement path for the competency of SDM tasks.

### 3.1. The Respondent Information Profile and Reliability Analysis

One hundred thirty-nine physicians’ questionnaires were gathered, and 118 were valid samples. The sociodemographic characteristics of the study participants (78 male, 40 female) were shown in Table 2. About 56% of the participants were aged less than 30 years. Approximately 80% of the clinicians worked at a medical center, and the remaining doctors were in the regional hospital. The SDM practice experience was less than six times in about 60% of the participants. Different levels of physicians included 34 attending physicians, 33 residents, and 51 doctors in post-graduate years. Cronbach’s Alpha was used to determine the aspects/criteria reliability for SI and AI. The reliability of the SI is 0.961, and the reliability of the AI is 0.974. The reliability of the SI and AI were higher than the suggested Cronbach’s Alpha (Cronbach = 0.7), so the SI and AI were highly consistent. The entire aspects’ reliability was 0.945, higher than the suggested Cronbach’s Alpha, so the entire aspects were highly consistent. The CE aspect’s reliability was 0.952, and the SP aspect’s reliability was 0.950. The CR aspect’s reliability was 0.954, and the JM aspect’s reliability was 0.964, higher than the suggested Cronbach’s Alpha, so these aspects’ items were highly consistent as indicated in Table 3.

### 3.2. The SAA-NRM Approach

Table 4 presents the evaluation obtained using SAA and NRM. The SAA analysis of this study is presented below: The first adoption step is to improve those aspects (i.e., SP and CE) falling into the third quadrant (L, L), indicating the low satisfaction index and low attention index. The second adoption step is to enhance those aspects (i.e., CR, JM) that fall into the first quadrant (H, H), showing the high satisfaction index and high attention index, as shown in Figure 2, left, Appendix A. Moreover, NRM analysis reveals that the CE and SP aspects are in the cause group, while the other two aspects, JM and CR, are in the effect group (Figure 2, right and Appendix A). The CE aspect influences the aspects of SP, JM and CR, and the SP aspect affects the aspects of JM and CR. Then, the JM aspect affects the CR aspect. Therefore, the best strategy for SDM competency development is to improve CE. Overall, Table 4 shows the four preferred development strategies. Preferred strategy A (situation maintaining) can be used in the CR and JM aspects. Preferred strategy C (sequentially strengthening) can apply to the CE and SP aspects.

### 3.3. Evaluate the Suited Improvement Paths Via Ranking the Standardized Satisfaction and Attention of the Aspects

We integrate the same preferred paths of the satisfaction and attention index via the rank of aspects, and establish the suited improvement paths. As illustrated in Appendix A, the standardized satisfaction (SS) of the CE, SP, CR, and JM aspects is −0.518, −1.137, 0.640, and 1.015, respectively. The standardized attention (SA) of the CE, SP, CR, and JM aspects is −1.039, −0.673, 0.899, and 0.813, respectively. In the suited improvement path analysis, the ranking of the SI is JM > CR > CE > SP, and the AI ranking is CR > JM > SP > CE. The study combines the improvement path of SI and AI, and there is no suited improvement path in the competence development of SDM tasks, as shown in Table 5. So, the decision-maker should use only the individual available improvement path for each aspect for the core competence of SDM tasks. In the SI, the CR aspect can be affected through the JM aspect in the second available improvement path (CE [3]→JM[1]→CR[2]). Then, the SP aspect can be affected through the CE aspect in the third available improvement path (CE[3]→SP[4]→CR[2]). The SP aspect can be affected through the CE aspect, and the CR aspect can be affected through JM aspect in the fourth available improvement path (CE[3]→SP[4]→JM[1]→CR[2]). In the AI, there is no available improvement path. Thus, no suited improvement path is obtainable after merging the improvement path of SI and AI.

### 3.4. Development Strategies and Suited Improvement Paths for Each Aspect

#### 3.4.1. The CE Aspect

With the CE aspect, the SAA-NRM analysis is shown in Table 6 and Figure 3. The net influence matrix in terms of the CE aspect is presented in Table 7. In the analysis of the SAA, the CE1 criterion is the satisfaction index less than the average satisfaction index (SI < 0) and the attention index greater than the average attention index (AI > 0). The CE1 criterion is within the second quadrant and should be used for preferred strategy B (direct strengthening). The CE3 criterion is the satisfaction index less than the average satisfaction index (SI < 0) and the attention index also less than the average attention index (AI < 0). The CE3 criterion is within the third quadrant and preferred strategy C (sequentially strengthening) should be applied. The CE2 and CE4 criteria are the satisfaction index greater than the average satisfaction index (SI > 0) and the attention index less than the average attention index (AI < 0). Therefore, these two criteria are within the fourth quadrant and should be enhanced by preferred strategy D (circumstance watching), as shown in Table 6 and Figure 3.

Based on the NRM analysis, the criteria of CE1 and CE4 are noted to show a positive net influence effect (d−r > 0). So the CE1 criteria only can be enhanced through itself, and the CE3 criterion can be improved through the CE1, CE4, and CE2 criteria, as presented in Figure 3, Table 6 and Table 7.

In the suited improvement path analysis, the ranking of the SI is CE2 > CE4 > CE3 CE1 and the ranking of the AI is CE1 > CE2 = CE4 > CE3, as indicated in Table 8. Therefore, the SAA-NRM technique combines the result of the SI and AI development paths, and there are three suited improvement paths (CE1→CE2→CE3; CE1→CE4→CE3; CE1→CE4→CE2→CE3) as shown in Table 8. The advantaged aspects/criteria can ameliorate the disadvantaged aspects/criteria.

#### 3.4.2. The SP Aspect

With the SP aspect, the SAA-NRM analysis is shown in Table 9 and Figure 4. The net influence matrix in terms of the SP aspect is shown in Table 10. In the SAA analysis, the SP1 and SP4 criteria are the satisfaction index less than the average satisfaction index (SI < 0) and the attention index greater than the average attention index (AI > 0). The SP2 and SP3 criteria are the satisfaction index less than the average satisfaction index (SI < 0) and the attention index also less than the average attention index (AI < 0). So, the SP1 and SP4 criteria should adopt preferred strategy B (direct strengthening), and preferred strategy C (sequentially strengthening) can be used for the SP2 and SP3 criteria. Based on the NRM analysis, the SP2 criterion is the positive net influence effects (d−r > 0). So the SP1 can be improved through the SP2 criterion, and the SP4 criterion can be improved by the SP2, SP1, and SP3 criteria. The SP2 criterion can be improved through itself, and the SP3 criterion can be enhanced through the SP2 and SP1 criteria, as presented in Figure 4, Table 9 and Table 10.

In the suited improvement path analysis, the SI ranking is SP4 > SP1 > SP2 > SP3, and the AI ranking is SP1 > SP4 > SP3 > SP2, as shown in Table 11. The study combines the same available development paths of SI and AI, and there is one suited improvement path (SP2→SP1→SP3→SP4).

#### 3.4.3. The CR Aspect

With the CR aspect, the SAA-NRM analysis is displayed in Table 12 and Figure 5. The net influence matrix in terms of the CR aspect is summarized in Table 13. In the analysis of SAA, the CR1, CR2, and CR4 criteria are the satisfaction index greater than the average satisfaction index (SI > 0) and the attention index greater than the average attention index (AI > 0). The CR3 criterion is the satisfaction index less than the average satisfaction index (SI < 0) and the attention index greater than the average attention index (AI > 0). So, the CR1, CR2, and CR4 criteria should adopt the preferred strategy A (situation maintaining), and preferred strategy B (direct strengthening) can be used for the CR3 criterion. Based on the NRM analysis, the CR3 criterion is the positive net influence effects (d−r > 0). Therefore, the CR3 criterion can be improved by itself, as presented in Figure 5, Table 12 and Table 13.

In the suited improvement path analysis, the ranking of the SI is CR4 > CE2 > CE1 > CR3, and the ranking of the AI is CR4 > CR2 > CR1 > CR3. The SI ranking and AI ranking are the same, so there is no suited improvement paths for the CR aspect as shown in Table 14.

#### 3.4.4. The JM aspect

With the JM aspect, the SAA-NRM analysis is shown in Table 15 and Figure 6. The net influence matrix in terms of the JM aspect is shown in Table 16. In the analysis of SAA, the JM1, JM2, and JM3 criteria are the satisfaction index greater than the average satisfaction index (SI > 0) and the attention index greater than the average attention index (AI > 0). The JM4 criterion is the satisfaction index greater than the average satisfaction index (SI > 0) and the attention index less than the average attention index (AI < 0). So, the JM1, JM2, and JM3 criteria can adopt the preferred strategy A (situation maintaining). The preferred strategy D (circumstance watching) can be used for the JM4 criterion. Based on the NRM analysis, the JM1 and JM2 criteria are the positive net influence effects (d−r > 0). Therefore, four criteria are the satisfaction index more than the average satisfaction index (SI > 0), so decision makers only kept them, as presented in Figure 6, Table 15 and Table 16.

Based on the suited improvement path analysis, the SI ranking is JM3 > JM1 > JM2 > JM4, and the AI ranking is JM3 > JM1 > JM2 > JM4, as shown in Table 17. So the SI available improvement paths and AI available improvement paths are same for the JM aspect. There are three suited improvement paths (JM1→JM2→JM3; JM1→JM4→JM3; JM1→JM2→JM4→JM3), as shown in Table 17.

## 4. Discussion

The current study assesses the status of satisfaction and attention through the SAA approach and determines the network relation structure via the NRM approach. The CE aspect (concept and evaluation) is the primary driving influencer of SDM competence development for full samples. One criterion for each of the four aspects is essential for improving physicians’ SDM competencies: CE1 (concept of SDM), SP2 (evidence-based medicine skills), CR3 (reply and teach), and JM1 (defining decision-making needs). This study has several findings about the development strategies and suited improvement paths for each aspect, which we discuss and compare our results with those of other studies below.

### 4.1. The CE Aspect

We find three suitable improvement paths for the CE aspect (CE1→CE2→CE3; CE1→CE4→CE3; CE1→CE4→CE2→CE3) as indicated in Table 8. The second suited improvement path is that the CE1 criterion influences the CE2 criterion, and the CE2 criterion affects the CE3 criterion. The concept of SDM is simple but not easy to implement. Many national education programs aim not only to train healthcare providers in SDM skills, but also to increase knowledge, attitudes, and awareness of SDM [44,75]. Constructing a framework that visually represents concepts may encourage learners to prefer visual learning. This improvement path finding suggested that teaching a precise framework that defines concepts of SDM could better motivate trainees to focus on learning the value and awareness of SDM [76]. Instructors can encourage their trainees to work on developing competencies in SDM by emphasizing the value and importance of SDM and highlighting the clinical conditions in which it is applicable [27].

The third suited improvement path is that the CE1 criterion influences the CE4 criterion, and the CE4 criterion affects the CE3 criterion. During the SDM process, patients can choose an option that incorporates their values and preferences into the content of evidence-based medicine. In addition to good patient-centered care, SDM endorses evidence-based practice by improving patient engagement with scientific information [77]. Evidence-based medicine (EBM) has been identified as “the careful, precise and judicious use of the best available evidence in decision-making regarding the care of each patient [78]”. The third path proposes that the concept of SDM is a fundamental precondition to the knowledge of EBM. Clinicians possess evidence-based knowledge that improves awareness at the time SDM begins.

The fourth suitable improvement path is the CE1 criterion influencing the CE4 criterion, and the CE4 criterion influencing the CE2 criterion, with the CE2 criterion influencing the CE3 criterion. Two conceptual frameworks to assist participants in understanding the components of SDM are the SHARE approach and the three-talk model [19,20,21]. Sackett et al. [78] have carefully embedded the EBM framework for clinical decision-making at the intersection of three practice pillars: clinical research evidence, clinical judgments, and patients’ values and preferences. Gagne’s nine events of instruction emphasize the importance of using frameworks—they can help learners to consolidate new knowledge through visual associations and permit teachers to quickly rethink concepts in the future to aid in reinforcing skills. [79]. The fourth improvement path finding indicated that highlighting the SDM concept can help integrate the knowledge of EBM, promote learners’ value and importance of SDM, and alert them to clinical scenarios that are suitable or not suitable for SDM.

### 4.2. The SP Aspect

There is one suitable improvement path for the SP aspect (SP2→SP1→SP3→SP4), as shown in Table 11. Hoffmann et al. [80] stated that SDM is “the intersection of patient-centered communication skills with EBM, in the highest quality of patient care”. Healthcare providers should identify and assess research evidence and practice patient-centered communication to support SDM during consultations. The approach of EBM involves five fundamental steps: ask, acquire, appraise, apply, and assess. The role of SDM could be noted within step 4 (apply) of the EBM process [29]. Using the best evidence to guide patient care receives little attention. SDM provides a way to help physicians offer care specific to each patient’s situation. Patient decision aids are evidence-based instruments created to assist patients in making preference-sensitive decisions. It can improve patient knowledge, reduce passivity in decision-making, increase the possibility that their choice aligns with their values, and activate patient engagement in decision-making [33,81]. So, this improvement path finding demonstrated that the practiced skill of EBM is an important prerequisite for conducting decision-making steps. The adequate utility of patient decision aids can improve patient engagement and decisional self-efficacy.

### 4.3. The CR Aspect

There is no suited improvement path available for the CR aspect, as shown in Table 14. SDM is an interactive communication strategy where physicians and patients collaborate on decision-making [75]. Communication is the physician’s responsibility. A good clinician-patient relationship is a way to reduce health inequities and facilitate social and health outcomes [82]. The Accreditation Council for Graduate Medical Education (ACGME) has recognized interpersonal and communication skills as one of the six core competencies required for physicians-in-training [83]. The disparities between physicians and patients, including gender, ethnicity, culture, and religion, can present bias in patient-physician communication [84]. The antecedent factors influencing communication and developed communication climates may directly affect how treatment options are mutually selected in the SDM process [85]. Röttele et al. observed that using generalizability theory to rate doctor-patient communication minimally indicates a doctor’s communication skills [86]. There are significant differences between doctors and patients regarding perceived SDM, effective and open communication, and satisfaction with communication. Doctors tend to overrate the extent to which they apply SDM. Therefore, doctors should engage their patients more in SDM, even if they believe they have done so adequately [86].

Health professionals must be encouraged to plan regular training in their use of SDM and to comprehend the importance of building trusting relationships with patients. Qualifying the clinician-patient relationship is critical to implementing SDM, so physicians should communicate effectively [87,88]. Verbal and non-verbal behavior and teach-back techniques were related to more patient-centered communication [37,49,53]. Interpersonal interactive communication is a significant and complex matter requiring continuous lifelong learning and improvement in daily practice for physicians. This study finds no suitable improvement path for the CR aspect. It is consistent with earlier findings suggesting that the impact of education programs on communication skills for teaching SDM to medical trainees is limited [4]. It means that all four criteria are crucial, and improvements are needed in all four criteria to enhance communication and interaction in the implementation of SDM.

### 4.4. The JM Aspect

There are three suitable improvement paths for the JM aspect (JM1→JM2→JM3; JM1→JM4→JM3; JM1→JM2→JM4→JM3), as shown in Table 17. Many scholars have introduced the procedural steps for implementing SDM [19,21,75,89,90,91]. Most of the process models suggested a step by step approach. Rusiecki et al. [75] proposed a circular seven-step model for SDM in which the ordering of steps is not fixed, allowing greater flexibility than previous linear models, that is more in various clinical settings. The revised three-talk model [20] emphasized fluid transition between talk steps with less explicit direction, recognizing that dialogues about decisions can be complicated and looping. Our study observed that the pathway for the JM1 criterion influencing the JM2 criterion, JM3 criterion, or JM4 criterion is followed by linear steps. The JM4 criterion influencing the JM3 criterion might be conducted in the circular step. In clinical practice, patients are often unable to make a final decision at the first discussion of SDM and may need to make a final decision at the next visit. It is also consistent with the findings that the JM4 criterion affects the JM3 criterion. The JM1 criterion states that clinicians should consider respecting patients’ values and preferences, integrate individual clinical situations, and determine tailored decisions. The JM3 criterion describes that physicians should invite patients and their families to participate, meaning patients’ preferred roles in decision-making is essential. A pilot study of an online case-based SDM training program observed two areas where clinicians lack confidence in their SDM practice: exploring patient values and identifying patients’ preferred roles in decision-making [92]. The JM1 criterion is the primary and dominant criterion, and the JM3 criterion is the primary criterion being dominated in the JM (Joint information and decision making) aspect. Physicians should learn to explore patient values, share information with the patient, make a follow-up plan, and co-participate in decision-making.

### 4.5. Study Comparison and Implications

Generally speaking, the use of decision support systems in health care is becoming increasingly common at many levels of decision-making [93]. For example, an incentive-based framework resting on a SWOT (strengths, weaknesses, opportunities, and threats) analysis would reinforce primary healthcare service, lessen friction between actors and entities, and finally guide future national and international primary healthcare reforms [94]. The analytic hierarchy process can be used to facilitate SDM, enhance clinician-patient communication, and determine priority options for colorectal cancer screening strategies. Liu et al. [8] applied the DEMATEL method to identify critical factors for SDM implementation. Jin Y et al. [95] proposed considering the interaction between the criteria in the SDM-questionnaire-nurse scale through NRM from the orthopedic nurse’s perspective. The SDM training programs mentioned in the past literature vary widely regarding course evaluation, training duration, training tools, teaching methods, participant reflection, advantages or abilities learned from SDM training, and facilitator or barriers to SDM implementation [6,24,27,96]. There is sparse evidence of which training programs are the most practical and which physicians’ SDM competencies are needed first. The current research found that the conception evaluation is the most important competency required to improve from the clinicians’ perspectives by the NRM method. The integrated SAA-NRM approach can aid clinicians in addressing gaps in needs, finding the most suitable path, and providing different SDM development strategies. Although there is no suitable improvement path among entire aspects via the SAA-NRM approach, it indicates that these four aspects are critical to developing clinicians’ SDM competencies.

### 4.6. Study Limitation and Future Studies

The design of the present study is not without limitations. First, the collected data in this study is restricted to one medical center and one regional hospital in southern Taiwan and may not apply to other medical institutions or countries. The generalizability of these findings in SDM research is unclear. Second, the participants in this study were enrolled using snowball sampling. Our results should therefore be interpreted with caution, as the sample may not represent the views of all Taiwanese doctors. Third, only three levels of physicians, namely, attending physicians, residents, and doctors in the post-graduate year, are involved in the research. The data supporting the SDM as presented in this study are limited to providing relevant information to medical doctors. Thus any suggestion for SDM implementation he would make can be biased. Other important stakeholders, such as healthcare providers, patients and their families, and policy-makers, can provide crucial information and insights related to developing the SDM process. Future studies could further investigate the viewpoints of different stakeholders. More survey subjects from more stakeholders may provide more insights and valuable recommendations for the success of SDM implementation. Besides, the current study’s analysis of physicians’ SDM competencies was not focused on specific diseases or challenging issues (e.g., patients’ inability to make decisions autonomously that requires dependence on family members or primary caregivers to make decisions). Future research could compare the disparity in the development strategy of physicians’ SDM competencies for particular clinical situations or different specialists, especially during the COVID-19 pandemic.

## 5. Conclusions

This study conducts an SAA-NRM approach to explore the critical factors of SDM competence and the interaction relation between aspects and further design appropriate development strategies. Precise quantification and network architecture will be helpful in increasing the consensus between the key elements in SDM competency development from physicians’ perspectives. Furthermore, physicians’ professionalism in performing SDM tasks may reduce patients’ uncertainty and anxiety, strengthen self-efficiency and trust, and diminish the barriers to SDM.

## Figures and Tables

**Figure 1 ijerph-19-13310-f001:**
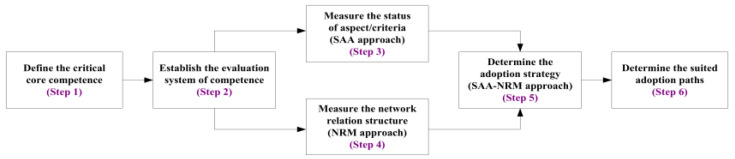
The SAA-NRM approach for core competence of SDM tasks.

**Figure 2 ijerph-19-13310-f002:**
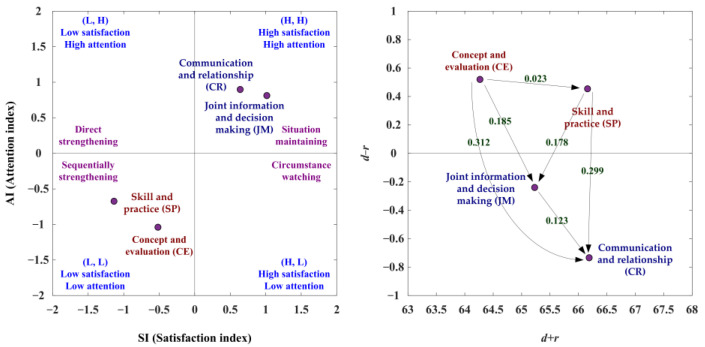
The SAA-NRM analysis of competence development of SDM tasks.

**Figure 3 ijerph-19-13310-f003:**
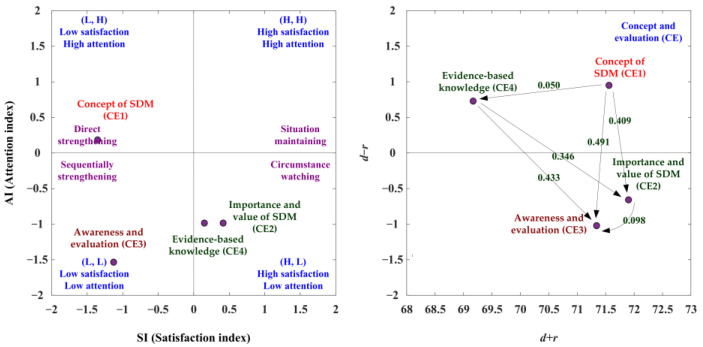
The SDM competency development map for the CE (concept and evaluation) aspect.

**Figure 4 ijerph-19-13310-f004:**
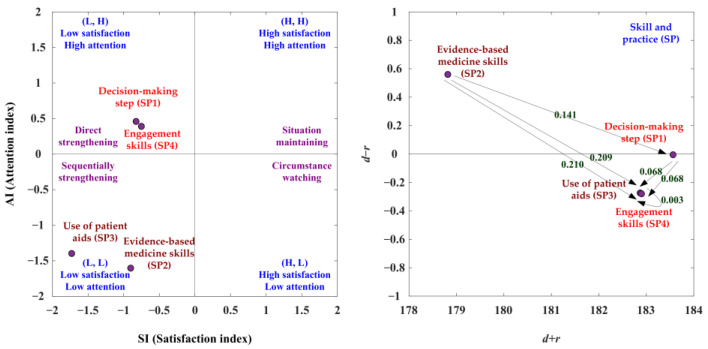
The SDM competency development map for the SP (skill and practice) aspect.

**Figure 5 ijerph-19-13310-f005:**
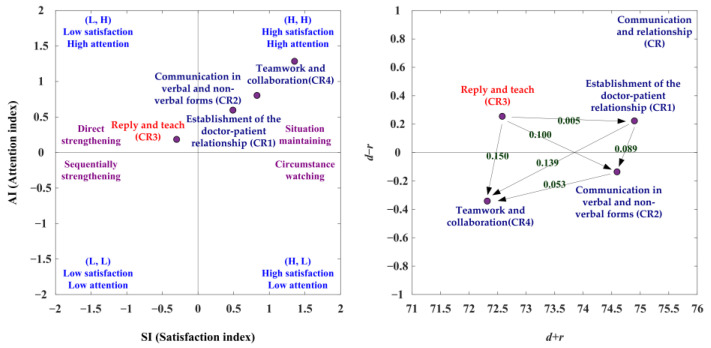
The SDM competency development map for communication and relationship (CR) aspect.

**Figure 6 ijerph-19-13310-f006:**
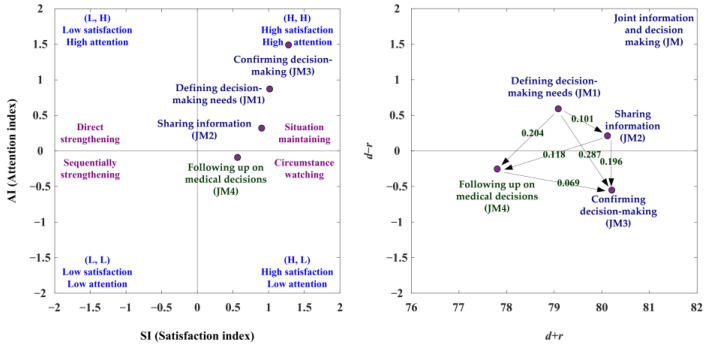
The development strategies map of the joint information and decision making (JM) aspect.

**Table 2 ijerph-19-13310-t002:** The sociodemographic characteristics of valid samples (N = 118).

Variables		n (%)
Gender	Male	78 (66.1)
	Female	40 (33.9)
Age	Less than 30 years	66 (55.9)
	31 to 40 years	30 (25.4)
	41 to 50 years	15 (12.7)
	Over 51 years	7 (5.9)
Workplace Institutions	Medical center	94 (79.7)
	Regional hospital	24 (20.3)
Different levels of physicians	Attending physicians	34 (28.8)
	Residents	33 (28)
	Doctors in post-graduate years	51 (43.2)
SDM practice experience	<6 times	70 (59.3)
	6–10 times	15 (15.3)
	>10 times	24 (20.3)

**Table 3 ijerph-19-13310-t003:** The analysis of reliability (Cronbach α ).

Items	Aspects/Criteria	Alpha	Result
Satisfaction index (SI)		0.961	High
Attention index (AI)		0.974	High
Entire aspects of the competence assessment		0.945	High
Each aspect	Concept and evaluation (CE)	0.952	High
	Skill and practice (SP)	0.950	High
	Communication and relationship (CR)	0.954	High
	Joint information and decision making (JM)	0.964	High

Note: Cronbach denotes Alpha *α*-value: *α* < 0.35 is low reliability, 0.35 < *α* < 0.70 is middle reliability, *α* ≥ 0.70 is high reliability.

**Table 4 ijerph-19-13310-t004:** The development strategy for competence development of SDM tasks.

Aspects	SAA			NRM			PS
	SI	AI	(SI, AI)	*D + r*	*d − r*	(R, D)	
Concept and evaluation (CE)	−0.518	−1.039	L, L	64.267	0.520	D (+, +)	C
Skill and practice (SP)	−1.137	−0.673	L, L	66.159	0.454	D (+, +)	C
Communication and relationship (CR)	0.640	0.899	H, H	66.188	−0.734	ID (+, −)	A
Joint information and decision making (JM)	1.015	0.813	H, H	65.229	−0.240	ID (+, −)	A

Notes: The preferred strategies (PS) include four types: preferred strategy A (situation maintaining), preferred strategy B (direct strengthening), preferred strategy C (sequentially strengthening), and preferred strategy D (circumstance watching).

**Table 5 ijerph-19-13310-t005:** The suited improvement paths of competence development of SDM tasks.

	SI (Satisfaction Index)	AI (Attention Index)
Rank	JM[1] > CR[2] > CE[3] > SP[4]	CR[1] > JM[2] > SP[3] > CE[4]
Available improvement paths	1. CE[3]→CR[2] {N} 2. CE[3]→JM[1]→CR[2] {Y} 3. CE[3]→SP[4]→CR[2] {Y} 4. CE[3]→SP[4]→JM[1]→CR[2] {Y}	1. CE[4]→CR[1] {N} 2. CE[4]→JM[2]→CR[1] {N} 3. CE[4]→SP[3]→CR[1] {N} 4. CE[4]→SP[3]→JM[2]→CR[1] {N}
Suited improvement paths	-	

**Table 6 ijerph-19-13310-t006:** The development strategies of the CE (concept and evaluation) aspect.

	SAA			NRM			PS
Criteria	SI	AI	(SI, AI)	*d + r*	*d − r*	(R, D)	
Concept of SDM (CE1)	−1.354	0.185	L, H	71.558	0.950	D (+, +)	B
Importance and value of SDM (CE2)	0.414	−0.984	H, L	71.902	−0.658	ID (+, −)	D
Awareness and evaluation (CE3)	−1.128	−1.534	L, L	71.341	−1.021	ID (+, −)	C
Evidence-based knowledge (CE4)	0.150	−0.984	H, L	69.169	0.729	ID (+, −)	D

Notes: The preferred strategies (PS) include four types: preferred strategy A (situation maintaining), preferred strategy B (direct strengthening), preferred strategy C (sequentially strengthening), and preferred strategy D (circumstance watching).

**Table 7 ijerph-19-13310-t007:** The net influence matrix of the CE (Concept and evaluation) aspect.

Criteria	CE1	CE2	CE3	CE4
Concept of SDM (CE1)	-			
Importance and value of SDM (CE2)	−0.409	-		
Awareness and evaluation (CE3)	−0.491	−0.098	-	
Evidence-based knowledge (CE4)	−0.050	0.346	0.433	-

**Table 8 ijerph-19-13310-t008:** The suited improvement paths of the CE (concept and evaluation) aspect.

	SI (Satisfaction Index)	AI (Attention Index)
Rank	CE2[1] > CE4[2] > CE3[3] > CE1[4]	CE1[1] > CE2[2] = CE4[2] > CE3[4]
Available improvement paths	1. CE1[4]→CE3[3]{N} 2. CE1[4]→CE2[1]→CE3[3] {Y} 3. CE1[4]→CE4[2]→CE3[3] {Y} 4. CE1[4]→CE4[2]→CE2[1]→CE3[3] {Y}	1. CE1[1]→CE3[4] {Y} 2. CE1[1]→CE2[2]→CE3[4] {Y} 3. CE1[1]→CE4[2]→CE3[4] {Y} 4. CE1[1]→CE4[2]= CE2[2]→CE3[4] {Y}
Suited improvement paths	2. CE1→CE2→CE3 3. CE1→CE4→CE3 4. CE1→CE4→CE2→CE3	

**Table 9 ijerph-19-13310-t009:** The development strategies of the SP (skill and practice) aspect.

	SAA			NRM			PS
Criteria	SI	AI	(SI, AI)	*d + r*	*d − r*	(R, D)	
Decision-making step (SP1)	−0.827	0.460	L, H	183.563	−0.005	ID (+, −)	B
Evidence-based medicine skills (SP2)	−0.902	−1.603	L, L	178.818	0.560	D (+, +)	C
Use of patient aids (SP3)	−1.730	−1.397	L, L	182.870	−0.274	ID (+, −)	C
Engagement skills (SP4)	−0.752	0.391	L, H	182.899	−0.280	ID (+, −)	B

Notes: The preferred strategies (PS) include four types: preferred strategy A (situation maintaining), preferred strategy B (direct strengthening), preferred strategy C (sequentially strengthening), and preferred strategy D (circumstance watching).

**Table 10 ijerph-19-13310-t010:** The net influence matrix of SP (skill and practice) aspect.

Criteria	SP1	SP2	SP3	SP4
Decision-making step (SP1)	-			
Evidence-based medicine skills (SP2)	0.141	-		
Use of patient aids (SP3)	−0.068	−0.209	-	
Engagement skills (SP4)	−0.068	−0.210	−0.003	-

**Table 11 ijerph-19-13310-t011:** The suited improvement paths of the SP (skill and practice) aspect.

	SI (Satisfaction Index)	AI (Attention Index)
Rank	SP4[1] > SP1[2] > SP2[3] > SP3[4]	SP1[1] > SP4[2] > SP3[3] > SP2[4]
Available improvement paths	1. SP2[3]→SP4[1] {N} 2. SP2[3]→SP3[4]→SP4[1] {Y} 3. SP2[3]→SP1[2]→SP4[1] {N} 4. SP2[3]→SP1[2]→SP3[4]→SP4[1] {Y}	1. SP2[4]→SP4[2] {N} 2. SP2[4]→SP3[3]→SP4[2] {N} 3. SP2[4]→SP1[1]→SP4[2] {Y} 4. SP2[4]→SP1[1]→SP3[3]→SP4[2] {Y}
Suited improvement paths	4. SP2→SP1→SP3→SP4	

**Table 12 ijerph-19-13310-t012:** The development strategies of the communication and relationship (CR) aspect.

	SAA			NRM			PS
Criteria	SI	AI	(SI, AI)	*d + r*	*d − r*	(R, D)	
Establishment of the doctor-patient relationship (CR1)	0.489	0.597	H, H	74.898	0.223	D (+, +)	A
Communication in verbal and non-verbal forms (CR2)	0.827	0.804	H, H	74.593	−0.136	ID (+, −)	A
Reply and teach (CR3)	−0.301	0.185	L, H	72.580	0.255	D (+, +)	B
Teamwork and collaboration(CR4)	1.354	1.285	H, H	72.317	−0.342	ID (+, −)	A

Notes: The preferred strategies (PS) include four types: preferred strategy A (situation maintaining), preferred strategy B (direct strengthening), preferred strategy C (sequentially strengthening), and preferred strategy D (circumstance watching).

**Table 13 ijerph-19-13310-t013:** The net influence matrix of the communication and relationship (CR) aspect.

Criteria	CR1	CR2	CR3	CR4
Establishment of the doctor-patient relationship (CR1)	-			
Communication in verbal and non-verbal forms (CR2)	−0.089	-		
Reply and teach (CR3)	0.005	0.100	-	
Teamwork and collaboration(CR4)	−0.139	−0.053	−0.150	-

**Table 14 ijerph-19-13310-t014:** The suited improvement paths of the communication and relationship (CR) aspect.

	SI (Satisfaction Index)	AI (Attention Index)
Rank	CR4[1] > CR2[2] > CR1[3] > CR3[4]	CR4[1] > CR2[2] > CR1[3] > CR3[4]
Available improvement paths	1. CR3[4]→CR4[1] {N} 2. CR3[4]→CR2[2]→CR4[1] {N} 3. CR3[4]→CR1[3]→CR4[1] {N} 4. CR3[4]→CR1[3]→CR2[2]→CR4[1] {N}	1. CR3[4]→CR4[1] {N} 2. CR3[4]→CR2[2]→CR4[1] {N} 3. CR3[4]→CR1[3]→CR4[1] {N} 4. CR3[4]→CR1[3]→CR2[2]→CR4[1] {N}
Suited improvement paths	-	

**Table 15 ijerph-19-13310-t015:** The development strategies of joint information and decision making (JM) aspect.

	SAA			NRM			PS
Criteria	SI	AI	(SI, AI)	*d + r*	*d − r*	(R, D)	
Defining decision-making needs (JM1)	1.015	0.873	H, H	79.085	0.592	D (+, +)	A
Sharing information (JM2)	0.902	0.322	H, H	80.118	0.213	D (+, +)	A
Confirming decision-making (JM3)	1.279	1.492	H, H	80.209	−0.551	ID (+, −)	A
Following up on medical decisions (JM4)	0.564	−0.090	H, L	77.802	−0.254	ID (+, −)	D

Notes: The preferred strategies (PS) include four types: preferred strategy A (situation maintaining), preferred strategy B (direct strengthening), preferred strategy C (sequentially strengthening), and preferred strategy D (circumstance watching).

**Table 16 ijerph-19-13310-t016:** The net influence matrix of the joint information and decision making (JM) aspect.

Criteria	JM1	JM2	JM3	JM4
Defining decision-making needs (JM1)	-			
Sharing information (JM2)	−0.101	-		
Confirming decision-making (JM3)	−0.287	−0.196	-	
Following up on medical decisions (JM4)	−0.204	−0.118	0.069	-

**Table 17 ijerph-19-13310-t017:** The suited improvement paths of the joint information and decision making (JM) aspect.

	SI (Satisfaction Index)	AI (Attention Index)
Rank	JM3[] >JM1[2] > JM2[3] > JM4[4]	JM3[1 >JM1[2] > JM2[3] > JM4[4]
Available improvement paths	1. JM1[2]→JM3[1] {N} 2. JM1[2]→JM2[3]→JM3[1] {Y} 3. JM1[2]→JM4[4]→JM3[1] {Y} 4. JM1[2]→JM2[3]→JM4[4]→JM3[ 1] {Y}	1. JM1[2]→JM3[1] {N} 2. JM1[2]→JM2[3]→JM3[1] {Y} 3. JM1[2]→JM4[4]→JM3[1] {Y} 4. JM1[2]→JM2[3]→JM4[4]→JM3[1] {Y}
Suited improvement path	2. JM1→JM2→JM3 3.JM1→JM4→JM3 4. JM1→JM2→JM4→JM3	

## Data Availability

The data presented in this study are available on reasonable request from the corresponding author.

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
