# Peer review of "Determining the Development Strategy and Suited Adoption Paths for the Core Competence of Shared Decision-Making Tasks through the SAA-NRM Approach"

_ijerph, 2022, doi:10.3390/ijerph192013310_

Round 1

Reviewer 1 Report

Abstract

Fine

Introduction

1)      The are some words which are not explained, as VALUE and PLACE.

Methods

1)     In section 2.1 it should be better explained how the literature review was performed, whether Bolean indicators were used, if not, the study may suffer from some bias.

2)     In section 2.6 you have on what basis did you decide the rank? should be better explained.

Results

1)     when using the acronyms SI, AI, etc., it does not matter to always use parentheses to repeat their meaning.

2)     To make the reading flow better, it would be appropriate whenever a "path" is mentioned (Table 15-18-21-24) to try to simplify the exposition, for example by eliminating the ranking exposition.

Discussion:

1.      I would remove the brackets from the acronyms.

2.      Table 25 can be removed.

Conclusion:

1)     Findings and Limitations should as the last paragraph of the discussion.

Author Response

Response to Reviewer 1 Comments

Point 1:

Abstract

Fine

Response 1: Thank you for your positive feedback.

Point 2:

Introduction:

1) The are some words which are not explained, as VALUE and PLACE.

Response 2: Thank you for your friendly reminder. We have added the full name of VALUE and PLACE (see line 50-53). We have a list of abbreviations at the end of the manuscript (see line 692-704).

Point 3:

Methods

1) In section 2.1 it should be better explained how the literature review was performed, whether Bolean indicators were used, if not, the study may suffer from some bias.

Response 3: Thank you for your suggestions. The literature search strategy of articles in our study was performed using PubMed and ScienceDirect database with Boolean operators to identify the SDM competency among physicians. This study used the keywords "shared decision making," "competency," "competencies," "knowledge," "skill," and "attitude" to review the literature and summarized the concepts of SDM, SDM assessment tools/scales, and essential elements in the SDM process. The study also adopts the MECE (Mutually Exclusive Collectively Exhaustive) concept to establish the evaluation system of service/decision process and check these indicators (aspects/criteria) through some field and practice experts (expert validity). Besides, the study analyzes the effectiveness and adequacy of the literature review for the evaluation system's indicators (aspects/criteria) through the reliability analysis (Cronbach Alpha). We have modified section 2.1 (see line 145-153).

Point 4:

2) In section 2.6 you have on what basis did you decide the rank? should be better explained.

Response 4: Thank you for pointing this out. Section 2.6 has been changed to Section 3.3 in the revised manuscript. The ranking of SI is based on the standardized satisfaction ranking of the CE, SP, CR, and JM aspects. The ranking of AI is based on standardized attention ranking of the CE, SP, CR, and JM aspects. As shown in Table S1, the standardized satisfaction (SS) of the CE, SP, CR, and JM aspects is -0.518, -1.137, 0.640, and 1.015, respectively. The ranking of SI is JM > CR > CE > SP. The standardized attention (SA) of the CE, SP, CR, and JM aspects is -1.039, -0.673, 0.899, and 0.813, respectively. The ranking of AI is CR > JM > SP > CE (see line 381-387).

Point 5:

Results

1) when using the acronyms SI, AI, etc., it does not matter to always use parentheses to repeat their meaning.

Response 5: Thank you for your comment. We only parenthetically clarify after the first abbreviation. The following abbreviations are not parenthetically explained, if possible.

Point 6:

2) To make the reading flow better, it would be appropriate whenever a "path" is mentioned (Table 15-18-21-24) to try to simplify the exposition, for example by eliminating the ranking exposition.

Response 6: Thank you for pointing this out. Table 15-18-21-24 has been changed to Table 8-11-14-17 in the revised manuscript. We tried simplifying the description and eliminated the ranking exposition in the revised manuscript, as the reviewer suggested.

Point 7:

Discussion:

1) I would remove the brackets from the acronyms.

Response 7: Thank you for your comment. We deleted the brackets from the acronyms.

Point 8:

2) Table 25 can be removed.

Response 8: Thank you for your comment. We removed Table 25 in the revised manuscript.

Point 9:

Conclusion:

1) Findings and Limitations should as the last paragraph of the discussion.

Response 9: Thank you for your comment. They have been placed in the last two paragraphs of the discussion section (see line 638-682).

Reviewer 2 Report

The manuscript reports a study on shared decision-making (SDM). SDM includes patients' opinions.

I have some concerns regarding the quality of the medical decisions made using the SDM framework as presented in this report.

1. Treatment suggestions are based on patient-centered (not centric) data

2. Opinions made by the patient during the SDM process are mainly based on suggestions made by medical doctors

The authors present a scenario where the patient regrets having opted for a treatment option suggested by the medical doctor because the medical doctor did not have an excellent SDM with the patient. I do not agree with this conclusion.  Furthermore, I disagree with the assertion that:

In personalized medicine, clinicians and patients utilize SDM to determine the best healthcare options among the available choices, considering the patient’s values, preferences, and clinical findings [8]. (Page 2).

I think there is some confusion here. Personalized medicine is based on patient-centric data instead of patient-centered data as used in the SDM framework presented here. Though, this assertion does not fit or support your study.

Omitting the data underlying the SDM as presented in this report, the authors make a good job. Though the entire study is biased since the data supporting the SDM as presented in this case study are limited to providing relevant information to the medical doctor, thus any treatment suggestion he would make can be biased. The SDM implies that the patient owns sufficient medical knowledge to understand the suggested treatment and to build a sustainable opinion.

Unfortunately, the study only considers the medical doctor training to support the SDM process "Medical training should provide the essential competencies a physician will need to perform well in the workplace" P.2. It will judicious to consider patient training, support patient medical literacy taking into account the education level of individuals.

Structure of the manuscript

The methodology section contains parts of the results.

The discussion needs to be improved. It sounds like a conclusion mixed with results

The tables and figures count is too high.

Author Response

Response to Reviewer 2 Comments

Point 1:

The manuscript reports a study on shared decision-making (SDM). SDM includes patients' opinions.

I have some concerns regarding the quality of the medical decisions made using the SDM framework as presented in this report.

  1. Treatment suggestions are based on patient-centered (not centric) data

  1. Opinions made by the patient during the SDM process are mainly based on suggestions made by medical doctors

The authors present a scenario where the patient regrets having opted for a treatment option suggested by the medical doctor because the medical doctor did not have an excellent SDM with the patient. I do not agree with this conclusion.  Furthermore, I disagree with the assertion that:

In personalized medicine, clinicians and patients utilize SDM to determine the best healthcare options among the available choices, considering the patient’s values, preferences, and clinical findings [8]. (Page 2).

I think there is some confusion here. Personalized medicine is based on patient-centric data instead of patient-centered data as used in the SDM framework presented here. Though, this assertion does not fit or support your study.

Response 1: The reviewer is correct. We apologize for the confusion we caused. To avoid ambiguity, we have removed the statement regarding reference 8. We have modified the introduction as the reviewer suggested.

Point 2:

Omitting the data underlying the SDM as presented in this report, the authors make a good job. Though the entire study is biased since the data supporting the SDM as presented in this case study are limited to providing relevant information to the medical doctor, thus any treatment suggestion he would make can be biased. The SDM implies that the patient owns sufficient medical knowledge to understand the suggested treatment and to build a sustainable opinion.

Response 2: Thank you for your suggestions. We have added the reviewer's comments to the limitations of our study for discussion (see line 669-671). We also have modified the introduction section as the reviewer suggested (see line 76-81).

Point 3:

Unfortunately, the study only considers the medical doctor training to support the SDM process "Medical training should provide the essential competencies a physician will need to perform well in the workplace" P.2. It will judicious to consider patient training, support patient medical literacy taking into account the education level of individuals.

Response 3: Thank you for your suggestions. We have added the reviewer's comments to the introduction section and removed inappropriate descriptions (see line 83-87).

Point 4:

Structure of the manuscript

The methodology section contains parts of the results.

Response 4: We apologize for the confusion we caused. To avoid ambiguity, we have rewritten our statements in the materials and methods section. The results have been presented uniformly in the results section.

Point 5:

The discussion needs to be improved. It sounds like a conclusion mixed with results

Response 5: Thank you for your nice reminder. The paper has been modified by removing a lot of redundant descriptions regarding the results in the discussion section. We have modified the discussion section in the revised manuscript.

Point 6:

The tables and figures count is too high.

Response 6: Thank you for your suggestions. The number of final tables has been reduced from 22 to 14. The number of final figures had declined from 7 to 5. Other forms are displayed in the Supplementary File.

Reviewer 3 Report

Dear authors,

I have enjoyed reading your manuscript, but I am afraid that its current version is not suitable for publication in the International Journal of 
Environmental Research and Public Health yet. In fact, my recommendation is that it needs a MAJOR REVISION.

However, fear not, since further details and some suggestions on how to improve your work are attached in my report.

I hope to receive a new revamped version of your study soon.

Yours sincerely,
Reviewer

Author Response

Response to Reviewer 3 Comments

Major comments

Point 1: On the one hand, the manuscript’s context and proposal are very well-defined. On the other hand, its motivation and contributions are not. In essence, it should be clear to the reader how and why the authors’ proposal is suited to address the identified knowledge gap: Are there any similar studies that can be comparable or serve as basis for the present one (even if not in Taiwan)? How does this study contribute to the existing literature?

Response 1: We thank the reviewer for this excellent suggestion. We have modified the introduction and discussion section (see line 112-116, 638-660).

Point 2: The authors mention that “After analyzing the existing literature and related works, we carried out expert interviews to identify the aspects of SDM competence”  to  come  up  with  4  aspects,  each  with  4  criteria.  What is the rationale behind the choice of these 4 by 4 dimensions? What is the weight that the literature and the experts had on their definition?

Responses 2: Thank you for your suggestion. Ospina et al [Ref 4] demonstrated that multiple definitions of SDM are available and vary between conceptual to more practical step-by-step approaches or frameworks where specific components are requirements of SDM. For this reason, we propose a framework inspired by the MECE (Mutually Exclusive Collectively Exhaustive) principle in decision science. The MECE scheme has three stages: measuring independence, importance, and completeness [Ref 22]. Thus, the current study adopts the MECE concept to establish the evaluation system of service/decision process and check these indicators (aspects/criteria) through some field and practice experts (expert validity). Besides, the study analyzes the effectiveness and adequacy of the literature review for the evaluation system's indicators (aspects/criteria) through the reliability analysis (Cronbach Alpha). Suppose the evaluation system cannot include these entire concepts. In that case, the study will increase new aspects/criteria, but the study will determine the whole aspects/criteria based on the MECE concept. We have added the source of the cited references and experts' viewpoints in Table 1.

Point 3: Obtaining such high Cronbach alphas is a phenomenal indicator of internal consistency. However, their values concern 118 valid responses from a sample that is unknown at the moment. Can the authors describe it and its possible imbalances and/or biases?

Response 3: Thank you for your comment. The sociodemographic characteristics of valid samples (78 male, 40 female) were shown in Table 2. About 56% of the participants were aged less than 30 years. Approximately 80% of the clinicians worked at a medical center, and the remaining doctors were in the regional hospital. The SDM practice experience was less than six times in about 60% of the participants. Different levels of physicians included 34 attending physicians, 33 residents, and 51 doctors in post-graduate years. We have modified the Result section (see line 335-340 and Table 2).

Point 4: Despite the popularity of the DEMATEL technique in several areas detailed in Subsection 2.4, why is it suitable for this particular context? Moreover, in its first  step,  how  mathematically  appropriate  is  it  to  make  computations  with ordinal scales? I ask this, because rather than a 0-4 discrete scale, one could have  used  a  1-5  or  0-40  scale  and  the  results  would  have  necessarily  been different due to the absence of a proportional interval between each number.

Response 4: Thank you for your suggestions. The DEMATEL technique was the systematic problem-solving tool. If the decision-makers can structure/systemic the decision problems, as shown in Figure 1, the DEMATEL technique can aid decision-makers in solving various field decision problems. The study adopted the 0-4 discrete scale based on the previous studies [Ref 50, 53, 56]. Still, the researchers can choose their preferred discrete scale for their questionnaire, and step 2 (compute the direct influence matrix) in the DEMATEL technique can normalize different discrete scales. Thus, the results are not different after calculation.

Point 5: Given the minimal difference of importance of influence between the four aspects, should not there have been conducted sensitivity analyses?

Response 5: Thank you for your suggestions. The degree of full influence (Table S7) can provide more information than the degree of direct influence (Table S4) and aid decision-makers in understanding the domination of aspect/criteria. So, the study based on the DEMATEL approach often adopts the degree of full influence to evaluate the influence/ domination of aspects/criteria.

Point 6: Although Section 3 is quite “heavy”, Section 4 should not follow its trend. Instead, it should be reformulated to provide clearer qualitative information to the reader on the results and derive the main policy implications. Moreover, have these results been shown to the experts and questionnaire responders that initially participated in the study?

Response 6: We apologize for the confusion we caused. To avoid ambiguity, we have rewritten our statements in the Materials and Methods section (Section 2). The results have been presented uniformly in the Results section (Section 3). The paper has been modified by removing a lot of redundant descriptions regarding the results in the Discussion section (Section 4). The study's experts know the findings, but the questionnaire responders do not. Our informed consent form states that questionnaire responders can contact the chief investigator at any time if they wish to know the study results. Our experts are members of the SDM Group at the Centre for Quality Management. They are committed to applying the findings of this study to the promotion and implementation of physician continuing education in our hospital.

Point 7: The manuscript is very well written. Still, I suggest the use of a specialised software for the occasional grammar typo.

Response 7: Thank you for your suggestion. ServiceScape Academic Editing edited this manuscript. We also used the Grammarly Premium software for checking grammar errors.

Minor comments

Point 8: What are VALUE and PLACE processes?

Response 8: Thank you for your nice reminder. We have added the full name of VALUE and PLACE (see line 50-53). We have a list of abbreviations at the end of the manuscript (see line 692-704).

Point 9: At last, the role played by incentives in several areas of the healthcare sector has been recently demonstrated by Pereira, Marques, & Figueira (2021). Fundamentally, I believe the authors’ study is aligned with and could benefit from some insights provided by this publication.

Response 9: Thank you for your excellent suggestion. We have added their important findings in the Discussion section (see line 640-644 and reference 90).

Once again, thank you very much for your comments and suggestions.

Thanks to the reviewers for the thoughtful and thorough review.

Hopefully, we have addressed all of your concerns.

Reviewer 4 Report

Dear respected authors, 

Normally as a reviewer, it is my duty to ask for some corrections in order to enrich your manuscript, but really I was impressed by the time that you spent on writing this research with valuable contributions based on my knowledge. 

It is well-written, integrated, and analyzed. Just conclusions and findings; need to be more explained and absolutely finding must come before conclusions. 

Author Response

Response to Reviewer 4 Comments

Dear respected authors, 

Normally as a reviewer, it is my duty to ask for some corrections in order to enrich your manuscript, but really I was impressed by the time that you spent on writing this research with valuable contributions based on my knowledge. 

Point 1:

It is well-written, integrated, and analyzed. Just conclusions and findings; need to be more explained and absolutely finding must come before conclusions. 

Response 1: Thank you for your suggestion. We have rewritten the discussion section to explain our conclusive findings and compare the results of other results (see line 521-529, 638-660).

Round 2

Reviewer 1 Report

The authors have accepted our suggestions and the article can be accepted

Author Response

Response to Reviewer 1 Comments

Point 1: The authors have accepted our suggestions and the article can be accepted.

Response 1: We appreciate the reviewer for your precious time reviewing our paper and providing valuable comments. Once again, thank you very much for your comments and suggestions.

Reviewer 2 Report

Based on revison provided, the manuscript can be accepted for publication.

I would like to recommend the authors to provide more reference on patient medical literacy and its link to SMD.

Author Response

Response to Reviewer 2 Comments

Point 1:

Based on revison provided, the manuscript can be accepted for publication.

I would like to recommend the authors to provide more reference on patient medical literacy and its link to SMD.

Response 1: Thank you for your comment. Medical literacy facilitates the acquisition, assessment, and application of information and promotes a patient's sense of control over decision-making and responsibility for meaningful SDM [Ref 11]. Patients' perceptions of inadequate health literacy prevent them from engaging in effective conversations with health providers [Ref 12]. Health literacy, especially the ability to access information and communicate interactively, plays a prominent role in implementing SDM with cancer patients [Ref 13]. We have modified the introduction section as the reviewer suggested (see line 83-87).

Once again, thank you very much for your comments and suggestions.

Thanks to the reviewer for the thoughtful and thorough review.

Hopefully, we have addressed all of your concerns.

Reviewer 3 Report

Dear authors,

I have enjoyed re-reading your manuscript and I am pleased to inform you that I believe it is now suitable for publication in the International Journal of Environmental Research and Public Health. You have responded quite positively to the reviewers' comments and made significant improvements to the study.

Good luck in the next steps of the production process.

Yours sincerely,

Reviewer

Author Response

Response to Reviewer 3 Comments

Dear authors,

I have enjoyed re-reading your manuscript and I am pleased to inform you that I believe it is now suitable for publication in the International Journal of Environmental Research and Public Health. You have responded quite positively to the reviewers' comments and made significant improvements to the study.

Good luck in the next steps of the production process.

Yours sincerely,

Reviewer

Response: We appreciate the reviewer for your precious time reviewing our paper and providing valuable comments. Once again, thank you very much for your comments and suggestions.
